# Emotion Recognition in Preterm and Full-Term School-Age Children

**DOI:** 10.3390/ijerph19116507

**Published:** 2022-05-26

**Authors:** Letizia Della Longa, Chiara Nosarti, Teresa Farroni

**Affiliations:** 1Developmental Psychology and Socialization Department, University of Padova, 35131 Padova, Italy; letizia.dellalonga@unipd.it; 2Department of Child and Adolescent Psychiatry, King’s College London, London SE5 8AF, UK; chiara.nosarti@kcl.ac.uk

**Keywords:** emotion recognition, socio-emotional functioning, facial expressions, development, preterm children

## Abstract

Children born preterm (<37 weeks’ gestation) show a specific vulnerability for socio-emotional difficulties, which may lead to an increased likelihood of developing behavioral and psychiatric problems in adolescence and adulthood. The accurate decoding of emotional signals from faces represents a fundamental prerequisite for early social interactions, allowing children to derive information about others’ feelings and intentions. The present study aims to explore possible differences between preterm and full-term children in the ability to detect emotional expressions, as well as possible relationships between this ability and socio-emotional skills and problem behaviors during everyday activities. We assessed 55 school-age children (*n* = 34 preterm and *n* = 21 full-term) with a cognitive battery that ensured comparable cognitive abilities between the two groups. Moreover, children were asked to identify emotional expressions from pictures of peers’ faces (Emotion Recognition Task). Finally, children’s emotional, social and behavioral outcomes were assessed with parent-reported questionnaires. The results revealed that preterm children were less accurate than full-term children in detecting positive emotional expressions and they showed poorer social and behavioral outcomes. Notably, correlational analyses showed a relationship between the ability to recognize emotional expressions and socio-emotional functioning. The present study highlights that early difficulties in decoding emotional signals from faces may be critically linked to emotional and behavioral regulation problems, with important implications for the development of social skills and effective interpersonal interactions.

## 1. Introduction

The ability to discriminate and interpret emotional signals of different facial expression is a main component of nonverbal communication. The recognition of facial expressions depends both on accurate visuo-perceptual processes and correct emotion categorization. The accurate decoding of facial features is an essential human ability that develops from the very beginning of life [1,2]. Newborns show preferential orienting and tracking for schematic face stimuli compared to scrambled faces [3,4]. This initial preference provides infants’ plastic cortical circuits with specific visual input, ensuring the appropriate specialization of later-developing cortical areas that support fast and accurate face processing (i.e., fusiform face area; [5]). Face recognition studies indicate that both featural (individual elements of the face, such as the mouth, eyes/brows) and configural processing (structural relationship between features) are involved in discriminating emotional facial expressions, and their relative contribution varies depending on the emotion [6,7]. A set of six basic emotion expressions has been identified, including: happiness, sadness, fear, anger, surprise, and disgust [8]. The ability to correctly identify these emotional categories from facial expressions is crucial to understanding others’ feelings and, therefore, regulating social behavior across the lifespan. Emotion-related brain circuits emerge early in life, supporting infants’ capacity to recognize the emotional signals of different facial expressions. Indeed, facial expressions have been shown to be a reliable source of information regarding emotional states and a primary mode for the communication of affect [9]. The accurate decoding of facial emotional cues represents a social skill with the potential of critically influencing social interactions, as learning to discriminate different emotions communicated by others is essential in order to respond with behaviors appropriate and adaptive to the specific social context [10]. Moreover, emotion recognition has been linked to the ability to perceive the impact of one’s own emotional expressions on others, resulting in emotional self-regulation [11]. Therefore, the processing of emotional expressions constitutes a key component of socio-emotional functioning, relating to the ability to successfully interact and communicate with others and to efficiently deal with one’s own and other’s emotions.

Although the ability to discriminate emotion expressions emerges early in life, full proficiency in processing subtle aspects gradually develops throughout childhood. Infants as young as seven months show sensitivity to different emotional expressions based only on unimodal visual information [12,13]. The early ability to discriminate changes in affective expressions critically impacts the development of social referencing, which involves the ability to use others’ affective cues, including facial expressions, for guiding exploratory behavior towards novel objects or events [14]. By preschool age, children are able to associate prototypical facial expressions with the corresponding label above chance level [15]. A variety of tasks have been used to assess the development of facial emotion recognition during childhood, showing a general improvement with age with some emotion-related differences [16,17,18]. Indeed, there is agreement in previous studies in suggesting that positive expressions, happiness in particular, are recognized earlier and more accurately than negative expressions [19,20]. Within negative expressions, there is evidence that sadness is accurately recognized earlier then fear, anger and disgust, which gradually develop in this order [17], while other findings suggest that the recognition of happiness, sadness and anger is highly accurate in early childhood, while the recognition of surprise, fear and disgust significantly improves over the late childhood [21]. It is important to notice that sex is another important factor that may play a role in the development of emotion recognition skills. The existing literature reports inconclusive results regarding sex differences in recognizing emotional facial expressions in childhood, with some studies reporting a small but consistent female advantage [21,22,23] and others showing little differences between females and males during late childhood [24], or showing no significant sex effects [25,26,27]. 

In contrast to children born at term, children born preterm (<37 weeks of gestation) have been shown to display difficulties in their ability to recognize emotional expressions, which may be linked to difficulties in socio-emotional processing and social interactions [28]. According to definitions by the World Health Organization, preterm birth can be further subdivided into moderate to late preterm (LP; 32 to 37 weeks); very preterm (VP; 28 to 32 weeks); and extremely preterm (EP; less than 28 weeks). Preterm birth is associated with both biological (e.g., structural and functional brain alterations) and environmental (e.g., perinatal stress and parenting vulnerabilities) risk factors that may lead to socio-emotional problems across the lifespan [29,30,31]. The immature nervous system is vulnerable to injury and altered development [32,33], specifically in areas that exhibit pronounced maturation during the last trimester of pregnancy when most preterm children are born, including those involved in processing emotions and social stimuli such as the limbic cortex [34]. Specifically, the brain network including the amygdala, the orbitofrontal cortex, the fusiform gyrus, and the superior temporal sulcus has been shown to mediate the capacity to efficiently detect and attend to emotional facial expressions [35]. Lower-level social-affective systems, including face perception and joint attention, are involved in processing the sensory input needed to inform, elaborate, and update internal models of social interactions, and thus critically shape the development of social capacities [36]. Indeed, the functional maturation of the so-called “social brain” is altered in infants with a family history of autism spectrum disorder, suggesting a key vulnerability for processing social information [37]. Notably, atypical patterns of brain maturation and connectivity in their “social brain” have been associated with socio-emotional behavioral outcomes in children who were born preterm [38,39,40,41]. 

According to a neuroconstructivism perspective [42], the interplay between the morphological and functional maturation of the brain and the influence of environmental factors plays a critical role in neurodevelopmental outcomes. Environmental factors during a period of rapid brain maturation and physiological vulnerability may critically affect socio-emotional developmental trajectories [43]. In particular, preterm newborns are exposed to early-life stress during their stay in the neonatal intensive care unit (NICU; [44,45]). This environment exposes newborns to sensory overload (e.g., bright lights, noises) and repetitive painful procedures (e.g., heel lancing, venipunctures, nasal suctioning), presenting infants with a variety of sensory stimuli that they are not developmentally prepared to handle [46,47]. Additionally, preterm newborns suffer from affective deprivation in terms of parental care, and this may influence infant development [44,45]. Currently, much emphasis has been put on multisensory interventions in the NICU that attempt to ameliorate sensory deficits derived from prematurity and improve infants’ neurodevelopment [47,48,49]. It is also crucially important to consider that parenting may act either as a protective factor against early-life stress or as an additional exacerbating risk factor for early-life stress, and a recent report indicates that parents need to feel more included in their newborn’s care and to have effective communication with medical staff [50]. Preterm birth and hospitalization are highly stressful experiences for parents [51] that may influence the formation of early parent–child bonds and later behavioral problems [52]. In Figure 1, we propose an integrative model that posits an interaction between the biological vulnerabilities and environmental factors which affect the development of socio-emotional functions [29,33].

The present study investigates preterm and full-term children’s ability to recognize emotional signals from faces, which is an important neurocognitive ability that supports socio-emotional functioning. We examine children’s ability to recognize different categories of emotional expressions and the relationship between this ability and socio-emotional skills and problem behaviors during daily activities, as reported by parents. Additionally, as children with different neurodevelopmental disorders often face difficulties in recognizing and understanding emotions, which has been related to cognitive skills [53], we include an evaluation of the children’s cognitive profiles. To what extent different cognitive functions are needed for emotion recognition in both typical and atypical development is still unclear [53]; thus, we account for different cognitive skills, including abstract reasoning, working memory, and attention, as preterm children tend to experience difficulties in multiple cognitive functions [54,55]. Irrespectively of cognitive profiles, we hypothesized that preterm children would show specific impairments in recognizing emotional expressions compared to full-term children. Moreover, we predicted that performance in the Emotion Recognition Task would correlate with the parent-reported socio-emotional difficulties. 

## 2. Materials and Methods

### 2.1. Participants 

The study was conducted at the Department of Developmental Psychology and Socialization of the University of Padova. A number of 55 Caucasian children between the ages of 6 and 11 years old were included in the study (34 children born preterm and 21 children born full-term). Participants in the preterm group were recruited from the association “Pulcino” in Padova, a center for children born preterm that provides support for premature infants and their families from the earliest stages of development to later childhood. Participants in the control group were recruited from the local community by contacting families of typically developing children in the same age range who had participated in previous studies. Participants’ characteristics are summarized in Table 1. For more detail regarding the preterm children’s neonatal clinical information, please see Appendix A. Note that the preterm group included children with different gestational ages. Parents gave written consent for their child’s participation after being informed about the whole procedure. The local Ethical Committee of Psychological Research (University of Padova) approved the study protocol.

### 2.2. Stimuli and Procedure

All participants completed a cognitive assessment that comprised the following: the Raven’s Colored Progressive Matrices (CPM [56]) to evaluate abstract reasoning, the digit span test forward and backwards (BVN 5-11 [57]) to estimate working memory span, the Attention Network Task (ANT [58]), which provides a measure of the three main components of attention (alerting, orienting and executive control), and a computerized version of the Berg Card Sorting Test (BCST [59]) to assess cognitive flexibility. 

Moreover, parents were asked to fill in some questionnaires to investigate children’s cognitive, emotional and behavioral functioning in everyday activities. These included the Strengths and Difficulties Questionnaire (SDQ [60]), which investigates the presence of behavioral and emotional difficulties as well as prosocial behaviors; the Emotion Regulation Checklist (ERC [61,62]), which investigates negativity and emotion regulation; the Temperament in Middle Childhood Questionnaire (TMCQ [63]), which investigates child temperament in the last 6 months; and the Behavioral Rating Inventory of Executive Function (BRIEF [64]), which investigates executive functioning. Finally, parents were asked to fill in the Parenting Stress Index (PSI [65]), which takes into account parenting distress, quality of the parent–child relationship, and child characteristics in order to measure parents’ stress levels. 

In order to examine the children’s ability to recognize and interpret facial emotions, we used the Emotion Recognition Task (ERT) built on Ekman and Friesen’s (1976) “pictures of facial affect” [66], adapted and validated by [67] with stimuli of children’s faces. The stimuli consisted of static images from the validated Dartmouth database of children’s faces [68]. Specifically, two models of Caucasian boys and two models of Caucasian girls were selected from the database in order to control for possible sex bias. For each model, six emotions were included: happiness, surprise, fear, anger, disgust, and sadness, plus neutral expressions. We grouped facial expressions into two broad categories based on their emotional valence (positive vs. negative), as previous evidence suggests that children initially categorize emotions as simply “feels good” vs. “feels bad” [69]. Moreover, in everyday life, facial expressions are rarely displayed at their maximum intensity, suggesting that the ability to detect subtle changes in facial expressions and to recognize less intense emotional expressions may be crucial for efficiently interacting with others [70]. Thus, for each emotion, two levels of intensity were created by morphing the neutral face with the emotional face of the same model with the software Phantamorph 5.4.4, 2014 and then correcting possible distortions occurring during the morphing process with Adobe Photoshop. The size of the facial stimuli was approximately 7.5 × 7.5 cm^2^. Each participant was presented with a total of 56 stimuli (2 intensity levels × 6 emotions × 4 models, plus 2 neutral expression × 4 models; Figure 2). The ERT took approximately 10 min to complete. Similar to [19], the task was made up of two testing blocks: the first block included facial expressions showing happiness, surprise, and fear, plus neutral faces, while the second block included facial expressions showing sadness, disgust, and anger, plus neutral faces. Neutral expressions were included in order to prevent children from thinking all faces expressed specific emotions. For each block, four boxes were placed in front of the participants, each with a cartoon face displaying one of the three emotions included in that block plus a neutral expression. Children were asked to identify and name each emotion. After that, the experimenter handed the facial expression images to participants one by one and asked them to examine the face and put the photograph into the box that corresponded to its emotional expression. ERT scoring allows the marking of correct and incorrect answers for both levels of intensity. The maximum score for each emotion was 8 (4 models × 2 intensity levels).

### 2.3. Statistical Analyses

All statistical analyses were performed using R 1.1.383, a software environment for statistical computing and graphics [71]. To test for group differences in cognitive performance and on the parent-reported questionnaires, we carried out *t*-tests. Before performing *t*-tests, we checked whether the assumption of normality of data distribution was met for each measure in both participant groups. The assumption was violated only in one SDQ subscale (peer problems); hence, a non-parametric test (Kruskal–Wallis rank sum test; see Table 3) was used in this instance. Regarding the assumption of the homogeneity of variance, we specifically used a Welch two-sample *t*-test, which is appropriate for testing the equality of two means from independent populations even when the variances are not equal. To analyze data from the Emotion Recognition Task, we used a mixed-effect model approach. The choice of using a mixed-effects model approach was determined by the possibility to take into account fixed effects, which are parameters associated with an entire population directly controlled by the researcher, and random effects, which are associated with individual experimental units randomly drawn from the population [72,73]. An Akaike information criterion (AIC) model comparison was used to compare sets of models fitted to the same data [74,75]. The model that produced the lowest AIC value was regarded as the most plausible [76]. More specifically, to carry out the generalized mixed models, we used “Glmer” from the “lme4” package [77]. In order to compute R-squared for the models, we used “r.squaredGLMM” from the MuMIn package [78], which takes into account the marginal R squared (associated with fixed effects) and the conditional one (associated with fixed effects plus random effects). For each model, we reported the marginal R squared. The *p* value was also calculated using the “lmerTest” package [79]. In addition, we were interested in exploring whether the ability to recognize emotional expressions was related to children’s socio-emotional functioning in everyday activities, as assessed in the parent-reported questionnaires. To this end, we carried out correlational analysis by using the “cor.test” function, which returns both the correlation coefficient and the significance level of the association between paired samples. Moreover, given the fact that the questionnaires may have been subject to parent-reporting bias, we decided to run partial correlations considering individual differences in levels of parent-related stress as reported in the subscale of Parental Distress from the PSI by using the function “partial.r”.

## 3. Results

### 3.1. Cognitive Assessment

Table 2 shows descriptive statistics and analysis for between-group comparisons for each cognitive test. We also report means and standard deviations for children born preterm subdivided into various prematurity classifications (see Appendix A). No significant group differences were found for CPM (t = 0.50, *p* = 0.619, Cohen’s d = 0.14), or digit span forward (t = 0.85, *p* = 0.400, Cohen’s d = 0.25) or backwards (t = 1.34, *p* = 0.190, Cohen’s d = 0.39). Likewise, the two groups showed no difference in attentional skills, as measured by the ANT: alerting (t = −0.67, *p* = 0.505, Cohen’s d = −0.19), orienting (t = −0.35, *p* = 0.730, Cohen’s d = −0.11) and executive control (t = −0.82, *p* = 0.417, Cohen’s d = −0.23). However, in a more complex task that assesses cognitive flexibility as a core executive function (BCST), a significant difference between groups emerged. The percentage of errors in preterm children was significantly higher than in the controls (t = −5.67, *p* < 0.001, Cohen’s d = −1.51). Preterm participants made more perseverative errors (t = −3.71, *p* < 0.001, Cohen’s d = −0.95) and non-perseverative errors (t = −2.08, *p* = 0.042, Cohen’s d = −0.53) than full-term participants. It is important to note that non-perseverative errors are common after a rule change as a new association must be identified using trial and error via feedback received after each card is sorted; however, perseverative errors highlight impaired cognitive flexibility [80].

### 3.2. Parent-Reported Questionnaires

Table 3 shows descriptive statistics and analysis for between-group comparisons for parent-reported questionnaires. We also report means and standard deviations for children born preterm subdivided into the various prematurity classifications (see Appendix A). Significant group differences were found in the Difficulties Score (t = −2.75, *p* = 0.009, Cohen’s d = −0.76) and Emotional Symptoms subscales (t = −2.15, *p* = 0.037, Cohen’s d = −0.56) of the SDQ. No significant group difference emerged in any subscale of the ERC, while difference between preterm and full-term children was found in the Negative Affectivity of the TMCQ (t = −2.43, *p* = 0.022, Cohen’s d = −0.78). Results from the BRIEF revealed significant group differences in the total score (t = −2.64, *p* = 0.012, Cohen’s d = −0.75) and in the Cognitive Control subscale (t = −2.99, *p* = 0.005, Cohen’s d = −0.84). Finally, preterm children’s parents reported significant higher Total Stress (t = −2.66, *p* = 0.012, Cohen’s d = −1.31), Dysfunctional Interaction (t = −2.66, *p* = 0.011, Cohen’s d = −0.74) and Difficult Child scores (t = −2.82, *p* = 0.008, Cohen’s d = −0.81).

### 3.3. Emotion Recognition Task

In order to analyze children’s ability to identify facial expressions of emotions in the Emotion Recognition Task, we considered the accuracy in the categorization of each facial stimulus presented (0 = error; 1 = correct response). Descriptive statistics are reported in Appendix A. We used a generalized mixed-effect model approach, testing eight nested generalized mixed-effects models. The null model (Model 0) included only the random effects of the participants. The first model (Model 1) included the effect of group (two levels; full-term vs. preterm children) as a fixed factor and participants as random factor in order to test the differences associated with preterm birth. Moreover, we were interested in investigating the possible effects related to the valence of the emotion and the level of intensity at which the emotion was expressed; therefore, we tested four additional models including the valence (two levels; positive vs. negative emotions; Model 2) and the level of intensity (pure vs. merged emotion expressions; Model 3) as fixed factors and their interaction with the group factor (Model 4 and 5). Furthermore, we wanted to control whether developmental changes influenced the recognition of emotions; therefore, we tested an additional model including age in months as a fixed factor (Model 6). Finally, we explored possible sex differences by testing an additional model including participant’s sex as a fixed factor (Model 7; Table 4).

The likelihood ratio test showed that Model 6 was the best at predicting accuracy in emotion recognition (Table 5). The model explained 20% of the variance (*p* = 0.009). There were main effects for valence (B = 1.58, SE = 0.19, *p* < 0.001), intensity (B = 1.37, SE = 0.16, *p* < 0.001) and age (B = 0.009, SE = 0.003, *p* = 0.007). Moreover, the interaction effect between group and valence (B = −0.46, SE = 0.23, *p* = 0.043) indicated that preterm children were less accurate in recognizing positive emotional expressions compared to full-term children (Figure 3).

We considered two different types of errors: failure to detect any emotion or misjudging the face as neutral (emotion omission), or mistaking an emotion for another (misidentification) [70]. In order to analyze the ERT errors, we tested six nested mixed-effects models. In each model, the number of errors was the dependent variable. The null model (Model 0) included only the random effect of the participants. The first (Model 1) included the effect of group (two levels; full-term vs. preterm children) as the fixed factor. As we were interested in investigating possible differences in the type of errors that the children made, we tested two additional models including the type of error (two levels; emotion omission vs. misidentification; Model 2) as a fixed factor and the interaction between group and type of error (Model 3). Finally, as we were interested in exploring whether developmental changes influenced the errors in recognizing emotional expressions, we tested two additional models including age in months as a fixed factor (Model 4) and the interaction between age and type of error (Model 5; Table 6).

The likelihood ratio test showed that the full model (Model 5) was the best at predicting errors in emotion recognition (Table 7). The model explained 20% of the variance (*p* = 0.014). The results revealed a main effect of group (B = 1.65, SE = 0.82, *p* = 0.047), suggesting that overall preterm children made more mistakes in identifying emotional expression compared to full-term children, and a main effect of type of error (B = 12.37, SE = 4.05, *p* = 0.003), showing that school-age children tended to commit more omissions than misidentifications. Moreover, the interaction effect between age and type of error (B = −1.44, SE = 1.16, *p* = 0.017) indicated that, with increasing age, the failure to detect any emotion (omission) gradually decreases, whereas the rate of misidentification remains constant across childhood (Figure 4).

### 3.4. Correlation between Emotion Recognition and Socio-Emotional Functioning

In order to explore individual differences in the ability to recognize emotional facial expression and the possible link with socio-emotional functioning in everyday activities, we performed correlational analyses between ERT performance and parent-reported questionnaires. We elected this exploratory approach, given the complexity of the experimental design (i.e., multiple measures) and the paucity of prior evidence on which to estimate expected effects, on each and every one of the indicators derived from the questionnaires. The results showed that children’s accuracy in identifying emotional expressions was negatively correlated with SDQ peer problems (r = −0.36, *p* = 0.013) and positively correlated with SDQ prosocial behavior (r = 0.31, *p* = 0.032). These relationships were differently modulated in the preterm and full-term groups (Figure 5), indicating a stronger negative correlation with SDQ peer problems in preterm children (r = −0.43, *p* = 0.017) and a stronger positive correlation with SDQ prosocial behavior in full-term children (r = 0.48, *p* = 0.050).

Notably the aforementioned correlations remained significant even when controlling for levels of parental stress. The results are summarized in Table 7.

Considering the ERC, there was a positive correlation between children’s ability to recognize emotional expression and emotion regulation (r = 0.31, *p* = 0.030), particularly in the group of children born preterm (r = 0.41, *p* = 0.025); however, while investigating the TMCQ, there was a negative correlation between emotion recognition and negative affectivity (r = −0.30, *p* = 0.043), particularly in the group of children born full-term (r = −0.42, *p* = 0.107; Figure 6). The same results were shown when running partial correlations controlling for the level of parental stress (see Table 8).

Finally, we performed correlation between the ERT and BRIEF scores. The results showed negative correlations between children’s ability to recognize emotion expression and behavioral functioning (r = −0.34, *p* = 0.017) as well as the emotional dimensions (r = −0.29, *p* = 0.043) of executive functioning, as reported by parents in everyday activities (Figure 7). These correlations remained significant when controlling for the level of parental stress (see Table 9).

## 4. Discussion

The present study aimed to explore the possible differences in facial emotion recognition between full-term and preterm children, and whether this early ability may be associated with socio-emotional functioning during everyday life. The results revealed that preterm children were less accurate than full-term children at detecting positive emotional expressions and they showed an increased risk of social and behavioral problems. Additionally, exploratory correlational analyses showed a relationship between the ability to recognize emotional expressions and socio-emotional functioning. In line with these results, previous studies suggest that preterm children show early signs reflecting social difficulties, including emotional and behavioral adjustment problems, which are manifested in relationship difficulties with peers and parents (i.e., low levels of positive play with peers and/or poor synchronous interactions with parents) [81,82,83]. During interpersonal exchanges, emotional facial expressions communicate important social signals, allowing the understanding, encoding and organization of information about the affective state of oneself and others. This indicates that emotion processing represents a cornerstone social cognitive skill for optimal social functioning and relates to appropriate social behavior [40]. Indeed, early impairment in recognizing emotional facial expressions may limit the possibility to use cues from another’s facial expression to adjust and efficiently communicate which, in turn, may lead to reduced opportunities for developing social skills. Thus, preterm children’s social difficulties may persist with age and potentially become exacerbated across adolescence and adulthood, interfering with their ability to establish social bonds and potentially leading to an increased risk of psychiatric disorders [29,84,85]. Although different studies have demonstrated emotion recognition difficulties and social impairment in preterm children, much less is known about the links between emotion processing and social functioning. 

Preterm and full-term children did not differ in terms of abstract reasoning, short-term (forward digit span) and working memory (backwards digit span), or attention skills. This general cognitive assessment was in fact designed in order to ensure comparable cognitive levels between the two groups. However, in line with previous studies [86], preterm children were found to have some difficulties in executive functioning, as indicated by a higher number of errors in the BCST. In line with this finding, parents of preterm children reported more difficulties in cognitive control during everyday activities compared to parents of full-term children, as measured by the BRIEF. Moreover, parents reported more emotional difficulties in the preterm group, as indicated by significantly higher SDQ Emotional Symptoms and TMCQ Negative Affect scores. Importantly, the PSI results suggest that, overall, parents of preterm children experience higher levels of stress compared to parents of full-term children, which is driven by the perception that their child presents with more self-regulatory problems than other children and by difficulties in establishing functional interactions with their child. Thus, in line with previous findings, a primary source of increased parental stress seems to be related to characteristics of children that fail to meet parental expectations, making parent–child interactions more challenging and less satisfactory [87].

The results of the Emotion Recognition Task suggest that the ability to recognize facial emotions develops with age, in line with previous results that suggest the existence of a long developmental pathway before children reach adult levels of accuracy and speed in emotion expression processing [88,89]. More specifically, considering the type of errors made by children, it appears that the number of misidentifications remains constant across childhood, while the number of omissions significantly decreases, suggesting that the ability to discriminate between subtle facial emotional and neutral expressions gradually improves across development. This result provides important information about the continued development of emotion processing across childhood; in particular, older children are more accurate at detecting emotional cues from faces, while younger children are more likely to misinterpret ambiguous facial signals as neutral expressions. In regard to possible sex differences, results from the current study do not support a female advantage in recognizing emotional faces, which has been found in some previous studies [22], supporting recent evidence that suggests an absence of sex differences [26,27]. Nevertheless, given the inconsistent findings in the existing literature, the possible emergence of a female advantage in facial expression recognition should be further investigated in light of the dynamic interactions between maturational and social factors [24]. Being able to decode subtle emotional cues is extremely important in everyday life when facial expressions are rarely displayed at their maximum intensity, and even small changes in expression may communicate different social signals that need to be correctly interpreted in order to appropriately modulate social behavior. Given the relevance of considering the intensity at which facial emotions are expressed, in the present study, we included two different levels of intensity. As expected, our results indicated that pure emotion expressions were better recognized than ambiguous facial expressions (i.e., merging between emotional and neutral expressions). Another factor that we took into consideration was the valence of emotional expressions. Numerous studies suggest that the developmental course of recognizing facial expressions depends on the type of emotion to be decoded, with happiness recognized earliest and with the greatest accuracy compared to other emotions [19,58,88]. Consistently with previous evidence, our results indicate that school-age children are more accurate at recognizing positive compared to negative facial expressions. Crucially, preterm children showed lower emotion recognition scores, in particular for positive emotions, suggesting that they may present a specific difficulty in decoding positive emotions with potential cascading effects on their capacity to appropriately engage in social interactions. This result is partially unexpected considering that, from a developmental perspective, positive emotions are recognized earlier and more accurately compared to negative emotions. Thus, one could expect that if preterm children still lag in decoding facial expressions compared to full-term children, they should have more difficulty detecting negative as opposed to positive emotional expression. However, early perceptual and social experiences have been shown to have a crucial role in shaping the perception of affective signals communicated by facial expression. Indeed, children who have experienced social environments that are marked by high levels of anger and hostility (e.g., maltreating families) begin to accurately recognize angry facial expressions earlier compared to children from non-maltreating families [90]. Although it is difficult to precisely measure the role of social experience on the organization of the developing perceptual system, it is possible to speculate that preterm children may have experienced perturbations in the frequency and content of emotional interactions (e.g., salient displays of negative affect) and they may have developed an increased perceptual sensitivity for variation in negative affective expressions, while showing specific difficulties in recognizing positive emotional expressions compared to full-term children. 

A pivotal aim of the present study was to investigate the possible relationships between emotion recognition and socio-emotional functioning in the everyday lives of preterm and full-term children. In order to achieve this objective, we investigated the association between children’s performance in the ERT and different parent-reported measures targeting children’s ability to regulate their behavior and affective state in an ecological social context. The observed associations should be interpreted with caution, given the multiple measures and the exploratory nature of these correlational analyses. The results showed that individual differences in the ability to recognize facial expressions correlated with SDQ pro-social behaviors in full-term children. Based on this result, it is possible to speculate that the ability to accurately decode facial emotional expressions allows typical-developing children to understand how others are feeling, thus guiding their social behavior. Conversely, in the preterm group, a negative correlation was noted between emotion recognition and SDQ peer problems, indicating that preterm children who are poor at recognizing emotional expressions may miss important social cues in interpersonal interactions and thus make errors in adjusting their behavior according to the social context, resulting in more difficulties in connecting with peers. In support of this interpretation, previous evidence suggests that the recognition of emotional expression is a crucial factor for the acquisition of social skills, and specific vulnerability in this early social-cognition ability has cascading effects on poor social and behavioral outcomes in children born prematurely [28,91]. Moreover, accuracy in decoding facial emotions is also believed to be associated with the ability to perceive the consequences of one’s own emotional expressions for others and consequently to learn to self-regulate emotional states [11]. Our results indicate a positive correlation between the recognition of emotional expressions and children’s emotional regulation, as assessed with the ERC, particularly in preterm children. Emotion regulation refers to the process of monitoring, evaluating and modifying emotional reactions and represents a core social skill that organizes interpersonal communication and social interactions [11]. Thus, individual variability in preterm children’ ability to identify emotional expression may be related to emotional self-regulation with important implications for socio-emotional development. The relationship between the ability to recognize emotional expressions and self-regulatory capacities was also seen in the analyses with the BRIEF. Specifically, in both groups of children, higher performance in the ERT correlated with fewer difficulties in behavioral and emotional functioning, which refers to the processes required for emotional self-regulation and goal-directed behavior, with important consequences for adaptive outcomes in everyday life [92]. 

A possible limitation of the current research is that the evaluation of children’s socio-emotional functioning was based solely on parent-reported questionnaires and, therefore, may have been subject to parent-reporting bias. However, by running partial correlational analyses, we controlled for parental stress, showing that the relationship between children’s ability to recognize emotional expressions and their emotional and behavioral outcomes persisted when the possible bias related to the level of parent-related stress was taken into account. In addition to parent-reported questionnaires, it would be interesting to investigate different aspects of children’s social competence, yet the lack of performance-based measures makes it difficult to design a reliable and standardized assessment of socio-emotional skills suitable for use with school-age children. Future works should address the need to extend and improve the limited selection of tools for the evaluation of children’s social and emotional functioning in an experimental setting. A second limit of the present study concerns the number of participants, as the sample size was determined by the number of families included in the database of the “Pulcino” association that agreed to participate in the study. The group of participants was rather small and a wide age range was considered in which relevant changes due both to maturation and environmental factors may occur. In order to overcome such variability across participants, we used standardized scores, except for the ERT, for which we adjusted for age in the statistical models. In order to better explore developmental changes, a larger sample of participants should be included in future studies. Moreover, the preterm group included children from extremely preterm to moderately preterm, which may have led to a wide heterogeneity of developmental trajectories. Future studies should include a large sample of preterm children balanced for their level of prematurity in order to better investigate possible differences related to variable gestational ages. In addition, the control group was matched for age, but not for sex. Given the fact that our results did not reveal a significant effect of sex on children’s ability to recognize emotional facial expressions, it is unlikely that the differences that emerged between preterm and full-term children were confounded by unbalanced sex distribution. Nevertheless, future investigations should attempt to match groups for age and sex. Finally, it is important to notice that the Emotion Recognition Task provides a valid behavioral measure of facial emotion recognition; however, it does not offer further information on the neural mechanisms that underpin the emotional processing of facial expressions. Future investigations should focus on the link between possible differences in the neural basis of emotional processing in preterm and full-term children and different socio-emotional developmental trajectories.

## 5. Conclusions

In conclusion, understanding the neurobiological risks and lingering adverse developmental outcomes of preterm birth is an important research field and will help shed light on the intervention programs designed to promote the better socio-emotional adjustment of preterm children. To this purpose, this study used many indicators to thoroughly examine the link between preterm children’s processing of emotional expressions and socio-emotional functioning. The results showed that preterm children present a specific vulnerability in decoding positive emotional expressions compared to full-term children, and they are at risk of social and behavioral problems. Notably, correlational analysis showed that children’s performance in the ERT was related to emotional self-regulation and to the ability of interacting with peers during everyday activities. However, more research is needed to identify the causal pathway underlying the processing of emotional expression and socio-emotional impairments. The findings of this study are particularly notable given that our preterm sample comprised children who appeared to have escaped major cognitive and attentional impairments. The results of the present research make an important contribution to improving our understanding of the socio-emotional functioning of preterm children, and contribute to the long-term aim of improving screening tools and clinical interventions used with populations at risk of impaired social functioning. 

## Figures and Tables

**Figure 1 ijerph-19-06507-f001:**
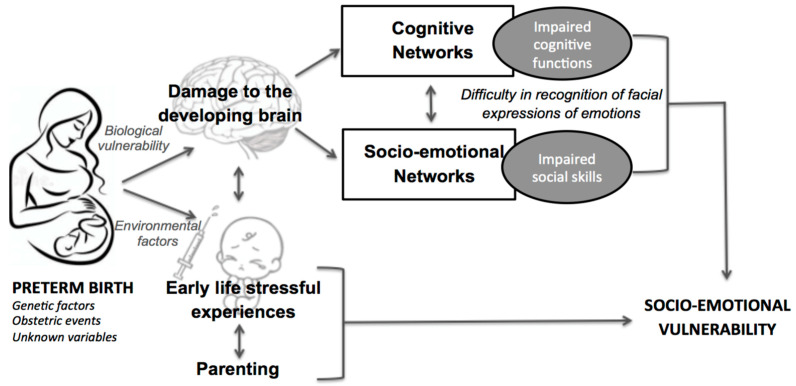
An integrative model showing the biological and environmental factors underlying preterm children’s socio-emotional vulnerability. Adapted from Montagna and Nosarti, 2016.

**Figure 2 ijerph-19-06507-f002:**
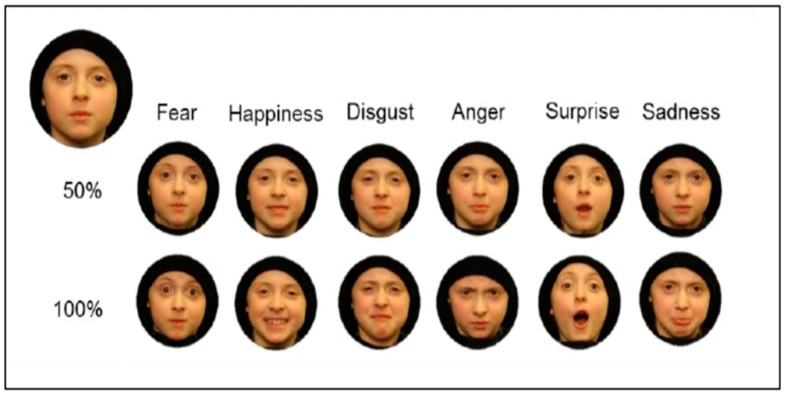
Example of the facial expression stimuli of the six emotions (happiness, surprise, fear, sadness, anger and disgust) at two levels of intensity (pure emotional expression and merged between neutral and emotional expression).

**Figure 3 ijerph-19-06507-f003:**
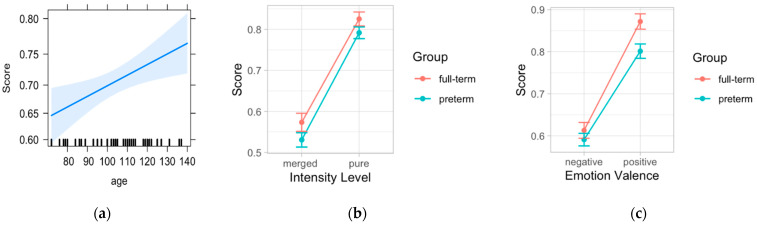
Plot of the main effects of age (**a**), intensity (**b**), and valence (**c**). The graph (**c**) shows also the interaction effect between group and valence.

**Figure 4 ijerph-19-06507-f004:**
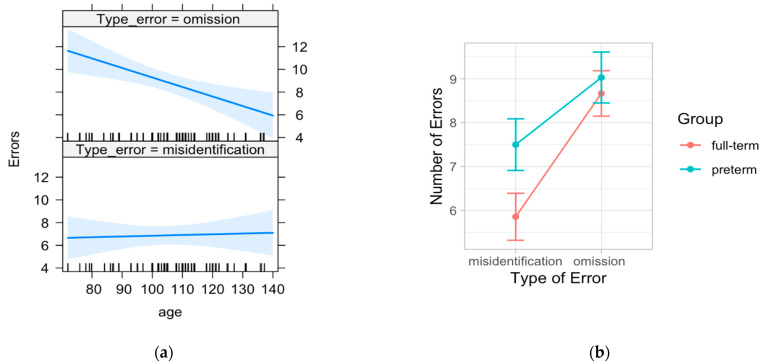
The left graph (**a**) shows of the interaction effect between age and type of error, while the right graph (**b**) shows the main effects of group and type of error.

**Figure 5 ijerph-19-06507-f005:**
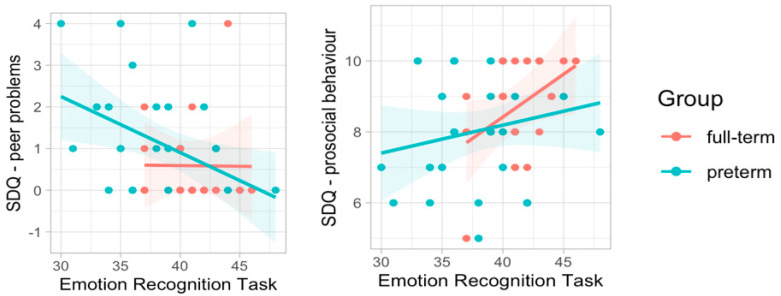
Correlations between ERT and the subscales of Peer Problems and Prosocial Behavior of the SDQ.

**Figure 6 ijerph-19-06507-f006:**
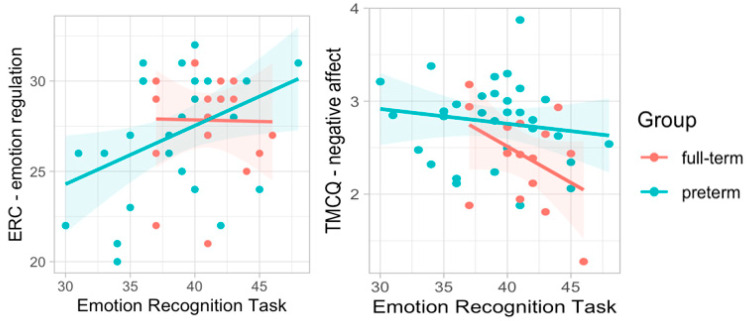
Correlations between ERT emotion recognition, ERC emotion regulation, and TMCQ negative affectivity.

**Figure 7 ijerph-19-06507-f007:**
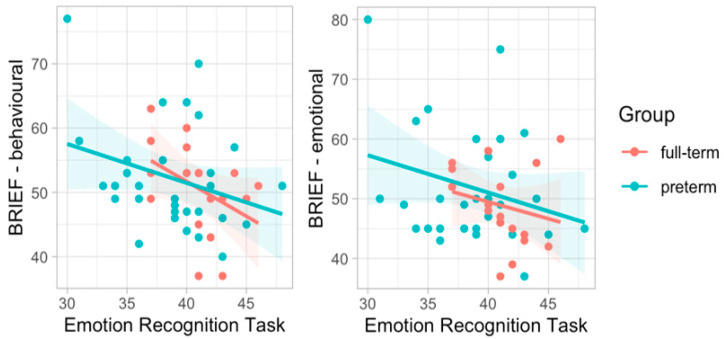
Correlations between ERT and subscales of behavioral and emotional functioning from the BRIEF.

**Table 1 ijerph-19-06507-t001:** Sample characteristics.

	Preterm Children	Full-Term Children
** *n* **	34 (19 M; 15 F)	21 (9 M; 12 F)
**Age (months)**	104.06 (15.43)	105.10 (15.94)
**Gestational age (weeks)**	29.91 (2.65) Late preterm (32–36 weeks) *n* = 8 Very preterm (28–31 weeks) *n* = 19 Extremely preterm (<28 weeks) *n* = 7	All > 37
**Birth weight (grams)**	1389.06 (556.49) Moderately preterm: 2041.25 Very preterm: 1350.11 Extremely preterm: 749.43	All > 2500

**Table 2 ijerph-19-06507-t002:** Descriptive statistics and analysis for between-group comparisons for each cognitive test.

	Full-Term Children	Preterm Children	Test forGroup Differences
**CPM**	0.86 (0.67)	0.77 (0.66)	t = 0.50, *p* = 0.619, d = 0.144
**Digit span forward**	0.06 (0.97)	−0.16 (0.81)	t = 0.85, *p* = 0.400, d = 0.247
**Digit span backwards**	0.66 (0.94)	0.34 (0.79)	t = 1.34, *p* = 0.190, d = 0.387
**Attention Network Task (ANT)**	Alerting: 29.72 (41.04)	A: 38.11 (50.62)	A: t = −0.67, *p* = 0.505, d = −0.190
	Orienting: 20.34 (63.84)	O: 26.25 (39.23)	O: t = −0.35, *p* = 0.730, d = −0.109
	Control: 34.50 (51.84)	C: 47.01 (61.96)	C: t = −0.82, *p* = 0.417, d = −0.227
**Berg Card Sorting Test (BCST)**	Errors: 27.02% (8.35) Perseverative Err: 14.25% (6.14) Non-Perseverative Err: 12.77% (7.44)	Err: 41.89% (10.81) Pers Err: 21.85% (7.94) Non Persev Err: 18.61% (5.68)	Err: t = −5.67, *p* < 0.001 Pers Err: t = −3.71, *p* < 0.001, d = −0.951 Non Pers Err: t = −2.08, *p* = 0.042, d = −0.532

**Table 3 ijerph-19-06507-t003:** Descriptive statistics and analysis for between-group comparisons for parent-reported questionnaires. Statistically significant differences between groups are marked with * (*p* < 0.05).

		Full-Term Children	Preterm Children	Test for Group Differences
**SDQ**	Difficulties Score:	5.00 (3.2)	8.03 (4.3)	t = −2.75, *p* = 0.009, * d = −0.763
	Prosocial Behavior:	8.65 (1.4)	8.10 (1.5)	t = 1.25, *p* = 0.219, d = 0.368
	Emotional Symptoms:	1.24 (1.1)	2.19 (1.9)	t = −2.15, *p* = 0.037, * d = −0.562
	Conduct Problems:	1.06 (1.0)	1.61 (1.2)	t = −1.74, *p* = 0.090, d = −0.492
	Hyperactivity:	2.12 (1.8)	3.16 (2.0)	t = −1.87, *p* = 0.069, d = −0.546
	Peer Problems:	0.59 (1.1)	1.06 (1.3)	H(1) = −2.369, *p* = 0.124, d = −0.386
**ERC**	Emotional Negativity:	25.53 (4.7)	27.11 (3.5)	t = −1.21, *p* = 0.238, d = −0.399
	Emotional Regulation:	27.83 (2.9)	27.10 (3.3)	t = 0.80, *p* = 0.426, d = 0.234
**TMCQ**	Surgency:	3.40 (0.5)	3.26 (0.4)	t = 0.89, *p* = 0.384, d = 0.292
	Effortful Control:	3.28 (0.9)	3.41 (0.5)	t = −0.55, *p* = 0.592, d = −0.199
	Negative Affect:	2.42 (0.5)	2.76 (0.4)	t = −2.43, *p* = 0.022, * d = −0.779
**BRIEF**	Total:	47.89 (7.0)	53.75 (8.3)	t = −2.64, *p* = 0.012, * d = −0.747
	Behavioral:	50.67 (7.0)	52.19 (8.3)	t = −0.69, *p* = 0.496, d = −0.195
	Emotional:	48.94 (6.6)	51.77 (9.6)	t = −1.21, *p* = 0.231, d = −0.327
	Cognitive:	47.00 (8.0)	54.71 (9.8)	t = −2.99, *p* = 0.005, * d = −0.839
**PSI**	Total stress:	57.83 (12.7)	67.30 (10.6)	t = −2.66, *p* = 0.012, * d = −0.830
	Parental distress:	20.44 (6.3)	22.53 (4.5)	t = −1.24, *p* = 0.226, d = −0.401
	Dysfunctional interaction:	18.39 (3.5)	21.57 (4.8)	t = −2.66, *p* = 0.011, * d = −0.733
	Difficult Child:	19.00 (4.8)	23.20 (5.4)	t = −2.82, *p* = 0.008, * d = −0.814
	Defensiveness Response:	12.56 (4.2)	14.10 (3.6)	t = −1.31, *p* = 0.201, d = −0.403

**Table 4 ijerph-19-06507-t004:** Comparison between models predicting accuracy in emotion recognition. Each model includes all the factors of the previous model plus an additional one. Note that smaller values of AIC indicate better fitting models.

Tested Models	Variables	AIC	Delta AIC	Marginal R^2^	χ^2^	*p*
Model 0	Random effect of participants	3330				
Model 1	+Group	3328	1.62	0.002	3.62	0.057
Model 2	+Valence	3179	148.6	0.087	150.64	<0.001
Model 3	+Intensity	2964	215.6	0.195	217.64	<0.001
Model 4	+Group × Valence	2962	2.02	0.200	4.02	0.045
Model 5	+Group × Intensity	2964	−1.80	0.201	0.20	0.655
Model 6	+Age	2959	4.92	0.205	6.92	0.009
Model 7	+Sex	2961	−1.96	0.205	0.04	0.836

**Table 5 ijerph-19-06507-t005:** Summary of the most plausible-fitting model predicting accuracy in emotion recognition.

Variables	B (SE)	Z Value	*p*
Group	−0.05 (0.14)	−0.36	0.717
Valence	1.58 (0.19)	8.38	<0.001
Intensity	1.37 (0.16)	8.82	<0.001
Age	0.009 (0.003)	2.71	0.007
Group × Valence	−0.46 (0.23)	−2.02	0.043
Group × Intensity	−0.09 (0.19)	−0.45	0.654

**Table 6 ijerph-19-06507-t006:** Comparison between models predicting errors in emotion recognition. Each model includes all the factors of the previous model plus an additional one. Note that smaller values of AIC indicate better fitting models.

Tested Models	Variables	AIC	Delta AIC	Marginal R^2^	χ^2^	*p*
Model 0	Random effect of Participants	576				
Model 1	+Group	576	1.43	0.023	2.53	0.112
Model 2	+Type of Error	566	10.13	0.118	11.58	<0.001
Model 3	+Group × Type of Error	567	1.34	0.127	1.17	0.280
Model 4	+Age	565	−4.02	0.157	4.22	0.040
Model 5	+Age × Type of Error	561	−0.99	0.199	6.03	0.014

**Table 7 ijerph-19-06507-t007:** Partial correlation between ERT and SDQ, controlling for PSI-Parental Distress.

Partial Correlation between SDQ Subscales and ERT Scores Controlling for PSI-Parental Distress			
	All Children ERT	Full-Term Children ERT	Preterm Children ERT
Difficulties Score	r = −0.28, *p* = 0.05	r = −0.05, *p* = 0.85	r = −0.24, *p* = 0.20
Prosocial Behavior	r = 0.31, *p* = 0.03	r = 0.48, *p* = 0.05	r = 0.22, *p* = 0.24
Emotional Symptoms	r = −0.16, *p* = 0.29	r = 0.11, *p* = 0.68	r = −0.13, *p* = 0.49
Conduct Problems	r = −0.11, *p* = 0.47	r = 0.30, *p* = 0.24	r = −0.14, *p* = 0.46
Hyperactivity	r = −0.17, *p* = 0.24	r = −0.35, *p* = 0.16	r = −0.02, *p* = 0.92
Peer Problems	r = −0.37, *p* = 0.01	r = 0.01, *p* = 0.97	r = −0.45, *p* = 0.01

**Table 8 ijerph-19-06507-t008:** Partial correlation between ERT and ERC and TMCQ, controlling for PSI-Parental Distress.

	All Children ERT	Full-Term Children ERT	Preterm Children ERT
**ERC** Emotional Negativity	r = −0.20, *p* = 0.16	r = −0.23, *p* = 0.38	r = −0.13, *p* = 0.48
**ERC** Emotional Regulation	r = 0.31, *p* = 0.03	r = −0.03, *p* = 0.90	r = 0.40, *p* = 0.03
**TMCQ** Surgency	r = −0.18, *p* = 0.24	r = −0.27, *p* = 0.31	r = −0.22, *p* = 0.23
**TMCQ** Effortful Control	r = 0.04, *p* = 0.79	r = −0.21, *p* = 0.42	r = 0.23, *p* = 0.22
**TMCQ** Negative Affect	r = −0.30, *p* = 0.04	r = −0.45, *p* = 0.08	r = −0.16, *p* = 0.38

**Table 9 ijerph-19-06507-t009:** Partial correlation between ERT and BRIEF controlling for PSI-Parental Distress.

	All Children ERT	Full-Term Children ERT	Preterm Children ERT
**BRIEF** Total	r = −0.21, *p* = 0.16	r = −0.26, *p* = 0.31	r = −0.07, *p* = 0.69
**BRIEF** Behavioral dimension	r = −0.34, *p* = 0.02	r = −0.43, *p* = 0.08	r = −0.30, *p* = 0.10
**BRIEF** Emotional dimension	r = −0.29, *p* = 0.04	r = −0.23, *p* = 0.37	r = −0.27, *p* = 0.14
**BRIEF** Cognitive dimension	r = −0.05, *p* = 0.72	r = −0.10, *p* = 0.72	r = 0.11, *p* = 0.53

## Data Availability

The data presented in this study are available on request from the corresponding author.

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
