# Peer review of "Emotion Recognition in Preterm and Full-Term School-Age Children"

_ijerph, 2022, doi:10.3390/ijerph19116507_

Round 1
Reviewer 1 Report
Thank you for the opportunity to review this article. Understanding the development of children born preterm has significant implications for early intervention and other support programs. The current manuscript provides an important foundation for future research. It addresses the development of emotion recognition with consideration to ER as a foundation for more complex socioemotional functioning that is important for quality of life.
While I found the manuscript to be well written overall, I have a few questions to be addressed.
First, I wondered if there had been consideration given to the vision of the participants. Based on the data from Table 1, many of the participants were born before 31 weeks gestation and below 1250gram therefore at high risk for retinopathy of prematurity (ROP). It may not be possible to detect lower grades of ROP during testing but even at lower grades it is likely to have an impact on the visual tasks presented. If data on ROP have not been collected, it would be a good idea to recontact families to obtain information on their child’s visual functioning.
I wondered how the full-term participants had been selected. It would be useful to have more detail, especially if there was an attempt to match on characteristics such as age. I can see that the mean age and standard deviation is similar, so I am guessing that this is what happened, but it would be good to have more details.
I noticed that the gender split is different with the full-term group having a higher proportion of females. This needs to be considered when interpreting group findings often show girls to perform better than boys on ER tasks [e.g. see Wang, Y., Hawk, S. T., Tang, Y., Schlegel, K., & Zou, H. (2019). Characteristics of emotion recognition ability among primary school children: Relationships with peer status and friendship quality. Child Indicators Research, 12(4), 1369-1388. – this is one of many studies to find girls do better in ER].
Comparison to the full-term participants is interesting, but I think a lot of readers would be interested in looking at some of the findings for the three different classifications of preterm. I appreciate that with low numbers and potentially high variability in age it might not be possible to do much more than descriptives, but I believe that could be helpful either within the article or as a set of tables in a supplementary file.
The introduction includes a lot of older citations and would benefit from including more recent research. This is important in all areas, but particularly for studies of preterm infants/children. For example, NICU procedures have changed considerably over the last two decades and perhaps some of the problems mentioned in the introduction were less pronounced when the participants in this study were born.
It would be helpful to highlight some of the difficulties related to gestational age and birthweight. The introduction, and the manuscript overall, has a tendency to consider all preterm children to be equally vulnerable rather than noting that there are substantial differences related to birthweight and gestational age.
I didn’t think Figure 1 was necessary. The text was adequate.
MDPI citation and referencing is required. Check the MDPI submission detail for information on citations and referencing. Make sure all cited references are in the reference list.
Line 11: remove ‘who were’
Line 451: form = from
Line 463, remove ‘to’
Author Response
RESPONSES TO REVIWER 1
1. First, I wondered if there had been consideration given to the vision of the participants. Based on the data from Table 1, many of the participants were born before 31 weeks gestation and below 1250gram therefore at high risk for retinopathy of prematurity (ROP). It may not be possible to detect lower grades of ROP during testing but even at lower grades it is likely to have an impact on the visual tasks presented. If data on ROP have not been collected, it would be a good idea to recontact families to obtain information on their child’s visual functioning.
R1. We thank the Reviewer for raising our attention to this important aspect. At the moment of testing, we run a small interview with parents in order to obtain information about possible complications at birth, their child developmental milestone stages (e.g. walk and language acquisition) and possible neurodevelopmental disorders. Regarding possible problems and complications at birth we specifically asked for neurological, respiratory, cariocirculatory, gastroenteric and sensory (including visual) problems. We included in Supplementary Materials a table with all information collected. In consideration to the risk of ROP, parents of two children reported respectively a retinopathy of stage 3, which was resolved with a laser operation within a month from birth, and a retinopathy of stage 2. Both of them were wearing glasses allowing them to compensate the sight. Looking at the individual performance, both children performed in line with the group of preterm children in recognizing the emotional expression and they correctly recognized the neutral expressions suggesting that it is unlikely the results emerged at the group level are related to an increased risk of retinopathy in preterm children.
Participant 1 with ROP: ERT score 69.64% of correct trials, 7/8 (87.5%) neutral expressions recognized
Participant 2 with ROP: ERT score 64.29% of correct trials, 8/8 (100%) neutral expressions recognized
Mean of the Preterm Group: ERT score 69.80 (SD= 7.41); recognition of neutral expressions 93.015 (SD= 11.18)
2. I wondered how the full-term participants had been selected. It would be useful to have more detail, especially if there was an attempt to match on characteristics such as age. I can see that the mean age and standard deviation is similar, so I am guessing that this is what happened, but it would be good to have more details.
R2. For the full-term group we recruited a convenient sample of children from an internal database of children who participated in previous studies. We were not able to collect a sample of control participants matched for both age and gender, but at least the two samples of participants were comparable in terms of age. We now included information about participant selection in the Participants section, as follow:
“Participants in the control group were recruited from the local community, contacting families of typically developing children in the same age range, who participated in previous studies.”
Furthermore, we comment on this aspect in the Discussion section, as follow:
“In addition, the control group was matched for age, but not for sex. Given the fact that our results did not reveal a significant effect of sex on children’s ability to recognize emotional facial expressions, it is unlikely that the differences emerged between preterm and full-term children may be confounded by unbalanced sex distribution. Nevertheless, future investigations should attempt to match groups for age and sex.”
3. I noticed that the gender split is different with the full-term group having a higher proportion of females. This needs to be considered when interpreting group findings often show girls to perform better than boys on ER tasks [e.g. see Wang, Y., Hawk, S. T., Tang, Y., Schlegel, K., & Zou, H. (2019). Characteristics of emotion recognition ability among primary school children: Relationships with peer status and friendship quality. Child Indicators Research, 12(4), 1369-1388. – this is one of many studies to find girls do better in ER].
R3. We thank the Reviewer for giving us the opportunity to further discuss our results in terms of possible gender differences. We did not found any effect of gender in our results, suggesting that the difference between groups is unlikely related to unbalanced gender proportion between groups. Indeed, descriptive statistics revealed that the performance was similar in males and females in both the preterm and the full term groups.
Percentage of correct responses in the ERT:
Control group: F= 73.36 (4.47), M = 73.81 (6.12)
Preterm group: F= 70.36 (5.22), M= 69.36 (8.89)
We also considered an additional model that includes the factor Gender and the Likelihood Ration Test shows that it did not increase the fit. However, given the importance of this aspect we now reported in the manuscript this additional model.
“Finally, we explored possible sex differences by testing an additional model including participant’s sex as a fixed factor (Model 7)”
|
Model 7 |
+ Sex |
2961 |
-1.96 |
.205 |
0.04 |
.836 |
Moreover we extended the Introduction and the Discussion sections.
Introduction: “It is important to notice that sex is another important factor that may play a role in the development of emotion recognition skills. The existing literature reports inconclusive results regarding sex differences in recognizing emotional facial expressions in childhood, with some studies reporting a small but consistent female advantage [21; 22; 23], and others showing little differences between females and males during late childhood [24] or showing no significant sex effects [25; 26; 27].”
Discussion:“In regard to possible sex differences, results from the current study do not support a female advantage in recognizing emotional faces, which has been found in some previous studies [22], supporting recent evidence that suggests an absence of sex differences [26, 27]. Nevertheless, given the inconsistent findings in the existing literature, the possible emergence of a female advantage in facial expression recognition should be further investigated in light of the dynamical interactions between maturational and social factors [24].”
See also comment above about the acknowledge of unbalanced gender proportion as a limitation of the current study.
4. Comparison to the full-term participants is interesting, but I think a lot of readers would be interested in looking at some of the findings for the three different classifications of preterm. I appreciate that with low numbers and potentially high variability in age it might not be possible to do much more than descriptives, but I believe that could be helpful either within the article or as a set of tables in a supplementary file.
R4. We thank the Reviewer for this suggestion and we now included in Supplementary Materials Tables with descriptive statistics specified for each sub-category of prematurity.
5. The introduction includes a lot of older citations and would benefit from including more recent research. This is important in all areas, but particularly for studies of preterm infants/children. For example, NICU procedures have changed considerably over the last two decades and perhaps some of the problems mentioned in the introduction were less pronounced when the participants in this study were born.
R5. We agree with the Reviewer about the importance of considering constant improvements in medical interventions as well as nursing practices that try to increase the benefit of the child while making a viable effort to decrease maternal stress levels. Currently, there is a great effort from multiple groups to use unisensory and multisensory stimulation in the NICU to ameliorate sensory deficits derived from prematurity (El-Metwally & Medina, 2020). However,the NICU represents per se a stressful and challenging environment and there is still urgency for medical staff in the NICU to become more aware of the needs and feeling of parents who are facing a very sensitive moment. Recent investigations indicate that mothers reported problems in the communication with medical staff, feeling of exclusion from the infant’s care and difficulties in adapt to NICU physical environment and regulations (Williams et al., 2018). Additional recent papers have been now cited in the introduction, such as El-Metwally & Medina, 2020; Filippa et al., 2020; Lammertink et al., 2021; Neel et al., 2019; Peralta-Carcelen et al., 2018; Williams et al., 2018, and we have now added them in the manuscipt:
“Currently, much emphasis has been put on multisensory interventions in the NICU that attempt to ameliorate sensory deficits derived from prematurity and improve infants’ neurodevelopment [47; 48; 49]. It isalso cruciallyimportant to consider that parenting may act either as a protective factor against early life stress or as additional exacerbating risk factor for early life stress and a recent report indicates that parents need to feel more included in their newborn’s care and to have effective communication with medical staff [50].”
6. It would be helpful to highlight some of the difficulties related to gestational age and birthweight. The introduction, and the manuscript overall, has a tendency to consider all preterm children to be equally vulnerable rather than noting that there are substantial differences related to birthweight and gestational age.
R6. We thank the Reviewer for this important comment and we have now included in the Introduction the classification of prematurity according to gestational age.
“According to definitions by the World Health Organization, preterm birth could be further subdivided into moderate to late preterm (LP; 32 to 37 weeks); very preterm (VP; 28 to 32 weeks); and extremely preterm (EP; less than 28 weeks).”
It is important to consider that preterm children show a high heterogeneity. Particular attention should be paid to EPs who are the most immature, tiniest and sickest infants. Along with increased survival in EP infants due to continue progress in perinatal care, the rates of infants considered at highest risk for later neurodevelopmental problems increased during the last decades (Morgan et al., 2022; Twihaar et al., 2018; Wilson-Costello et al., 2005). However, even in the case of LP children, who typically avoid major clinical complication and cognitive deficits, significant neuropsychological and behavioral difficulties might emerge during school age or later (Mento & Nosarti, 2015). Thus, in our study we decide to include all the range of preterm born children to investigate the effect of the interruption of prenatal brain development as a predictors of different neurodevelopmental trajectories that support social and behavioral functioning in everyday life.
7. I didn’t think Figure 1 was necessary. The text was adequate.
R/. We appreciate that the Reviewer found the text exhaustive, however we believe that the image could help to summarize the main points of the theoretical framework and we suggest to keep it in the text. We have now specify “In Figure 1 we propose an integrative model that posits an interaction betweenbiological vulnerabilities and environmental factors, which affect the development of socio-emotional functions.” Nevertheless we are prone to delete the image if the Reviewers found it redundant.
8. MDPI citation and referencing is required. Check the MDPI submission detail for information on citations and referencing. Make sure all cited references are in the reference list.
R8. Citations have been amended following MDPI requirements.
9. Line 11: remove ‘who were’; Line 451: form = from; Line 463, remove ‘to’
R9. We thank the Reviewer for noticing these errors; we have now removed/corrected these words.

Reviewer 2 Report
This is an interesting manuscript on children born preterm (<37 weeks’ gestation) show a specific vulnerability. More precisely, y to detect emotional expressions and the possible relationship with socio-emotional skills and problem behaviors during everyday activities. It is a nice work but I would like to suggest some points to authors before acceptance:
- Introduction:
-Accurate decoding of facial features is a tricky subject. I would mention that by citing some literature such as:
Moret-Tatay, C., Fortea, I. B., & Sevilla, M. D. G. (2020). Challenges and insights for the visual system: Are face and word recognition two sides of the same coin?. Journal of Neurolinguistics, 56, 100941.
-When talking about emotions, I expected at some point the specification of the types of emotions (e.g., Ekman's classification).
2. Method
More work should be done on the congruence of the hypothesis of the work with the statistical strategy. It is not clear, for example, to test models
3. Results:
-Are the statistical assumptions met for each proposed test?
-The effect size should be included in the t-tests or similar.
I see promise in this work, and I hope these comments could be of interest.
Author Response
RESPONSES TO REVIEWER 2
1. Introduction: Accurate decoding of facial features is a tricky subject. I would mention that by citing some literature such as: Moret-Tatay, C., Fortea, I. B., & Sevilla, M. D. G. (2020). Challenges and insights for the visual system: Are face and word recognition two sides of the same coin?. Journal of Neurolinguistics, 56, 100941.
R1. We thank the Reviewer for this suggestion. We have now added at the very beginning of the Introduction section an overview of facial processing in a developmental perspective, as follow:
“The ability to discriminate and interpretemotional signals of different facial expression is a main component of nonverbal communication. Recognition of facial expressions depends both on accurate visuo-perceptual processes and correct emotion categorization. Accurate decoding of facial features is an essential human ability that develops from the very beginning of life [1; 2]. Newborns show preferential orienting and tracking for schematic face stimuli compared to scrambled faces [3; 4]. This initial preference provides infants’ plastic cortical circuits with specific visual input, ensuring appropriate specialization of later developing cortical areas that support fast and accurate face processing (i.e. fusiform face area; [5]). Face recognition studies indicate that both featural (individual elements of the face, such as mouth, eyes/browns) and configural processing (structural relationship between features) are involved in discriminating emotional facial expressions and their relative contribution varies depending on the emotion [6; 7].”
2. When talking about emotions, I expected at some point the specification of the types of emotions (e.g., Ekman's classification).
R2. In accordance to the Reviewer’s comment we included Ekman’s classification of emotion, as follow:
“A set of six basic emotion expressions has been identified, including: happiness, sadness, fear, anger, surprise and disgust [8]. The ability to correctly identify theseemotional categories from facial expressions is crucial to understand others’ feelings and therefore regulate social behavior across the lifespan”.
3. Method: More work should be done on the congruence of the hypothesis of the work with the statistical strategy. It is not clear, for example, to test models
R3. The choice of using mixed-effect models was explained in the paragraph 2.3. Statistical analyses “To analyze data from the emotion recognition task we used a mixed-effect model approach. The choice of using a mixed-effects model approach was determined by the possibility to take into account fixed effects, which are parameters associated with an entire population as they are directly controlled by the researcher, and random effects, which are associated with individual experimental units randomly drawn from population (Baayen et al., 2008; Gelman & Hill, 2007).”
While the choice of including the different factors was explained in the paragraph 3.3. Emotion Recognition Task: “The null model (Model 0) included only the random effect of Participants. The first model (Model 1) included the effect of Group (2 levels; full-term vs preterm children) as fixed factor and Participants as random factor, in order to test differences associated with preterm birth. Moreover, we were interested in investigating possible effects related to the valence of the emotion and the level of intensity at which the emotion was expressed; therefore, we tested four additional models including the Valence (two levels; positive vs negative emotions; Model 2), the Level of intensity (pure vs merged emotion expressions; Model 3) as fixed factors and their interaction with the factor Group (Model 4 and 5). Furthermore, we wanted to control whether developmental changes may influence the recognition of emotions, therefore we tested an additional model including age in months as a fixed factor (Model 6). Finally, we wanted to consider possible gender differences, thus we test an additional model including participant’s gender as a fixed factor (Model 7; Table 4).”
All the factors that we included in our analyses, have been explained in the Introduction: “Therefore, this study aims to examine the ability to recognize different categories of emotional expressions in a group of preterm and full-term school-age children and its relationship with socio-emotional skills and problem behaviors during everyday activities, as reported by parents (parent-reported questionnaires describing children’s emotion and behavioral regulation). Specifically, we used pictures of peers’ faces expressing different emotions and we take into account the valence of the emotional expression (positive vs negative), as well as the level of intensity at which each emotion was expressed. We grouped facial expressions into two broad categories based on their emotional valence, as previous evidence suggests that initially children simply categorize emotions as “feels good” vs “feels bad” (Widen, 2013). Moreover, in everyday life, facial expressions are rarely displayed at their maximum intensity, suggesting that the ability to detect subtle changes in facial expressions and to recognize less intense emotional expressions may be a crucial for efficiently interact with others (Gao & Maurer, 2009). Thus, it is essential to take both qualitative valence and emotion intensity into account when studying children’s ability to recognize facial expressions as a prerequisite of interpersonal social interactions. […] Irrespectively of cognitive profiles, we hypothesized that preterm children will show specific impairments in recognizing emotional expressions, in particular at lower intensity levels, compared to full term children. Moreover we predicted that performance in the emotion recognition task would correlate with parent reported socio-emotional difficulties.” We also discussed the importance of considering developmental changes across different ages and possible gender differences in emotion recognition “The ability to discriminate emotion expressions emerges early in life, but full proficiency in processing subtle aspects of facial expression gradually develops throughout childhood […]” and “It is important to notice that gender is another important factor that may play a role in the development of emotion recognition skills. Existing literature is discrepant about gender differences across childhood with some studies that found a small but consistent female advantage in recognizing emotional facial expressions (Lawrence et al., 2015; McClure, 2000; Wang et al, 2019), while other studies point to a different developmental trend in females and males that tend to close the gender gap during late childhood (Mancini et al., 2013) or did not revealed a significant effect of gender (Calvo et al., 2008; Garcia & Tully, 2020; Romani-Sponchiado et al., 2022).”
However, if the Reviewer is still not convinced about the congruence of statistical analyses with the hypothesis we can better specify our choices. We kindly ask the Reviewer to specify which particular factor included in the models is not clear enough.
4. Results: Are the statistical assumptions met for each proposed test?
R4. We thank the Reviewer for giving us the opportunity to further explain this important point. For each measure we check whether the assumption of normality of data distribution was met for both groups of participants (preterm and full term children). Only in one subscale of SDQ we found that collected date were not normally distributed, thus in this case we used Kruskal-Wallis rank sum test (Kruskal-Wallis chi-squared = 2.3692, df = 1, p-value = 0.1237; see Table 3).
One-sample Kolmogorov-Smirnov test
Preterm: D = 0.247, p-value = 0.045
Full-term: D = 0.406, p-value = 0.007
We also check for homogeneity of variance between groups, which was met in most of the measures but not all of them. However, in our analyses we used Welch Two Sample t-tests, which is appropriate for testing the equality of two means from independent populations even when the variances are note equal.
We have now included these details in the Paragraph 2.3. Statistical analyses as follow:
“Before performing t-tests, we checked whether the assumption of normality of data distribution was met for each measure in both participant groups. The assumption was violated only in one SDQ subscale (peer problems), hence a non-parametric test (Kruskal-Wallis rank sum test; see Table 3) was used in this instance. Regarding the assumption of homogeneity of variance, we specifically used Welch Two sample t-test, which is appropriate for testing the equality of two means from independent populations even when the variances are note equal.”
5. The effect size should be included in the t-tests or similar.
R5. We already included in the text the Cohen’s D for t-tests that showed a significant effect. However, in following with the Reviewer’s comment, we have now also added Cohen’s D values for each t-test (see Table 2 and Table 3).
6. I see promise in this work, and I hope these comments could be of interest
R6. Indeed, and we thank the reviewer for the useful comments
